# The Emotional Impact of a Cancer Diagnosis: A Qualitative Study of Adolescent and Young Adult Experience

**DOI:** 10.3390/cancers16071332

**Published:** 2024-03-29

**Authors:** Luke Hughes, Rachel M. Taylor, Angharad E. Beckett, Oana C. Lindner, Adam Martin, Joanne McCulloch, Sue Morgan, Louise Soanes, Rizwana Uddin, Dan P. Stark

**Affiliations:** 1Cancer Clinical Trials Unit, University College London Hospitals NHS Foundation Trust, London NW1 2PG, UK; luke.hughes3@nhs.net; 2Centre for Nurse, Midwife and AHP Led Research (CNMAR), University College London Hospitals NHS Foundation Trust, London NW1 2PG, UK; rtaylor13@nhs.net; 3Department of Targeted Intervention, University College London, London WC1E 6BT, UK; 4School of Sociology and Social Policy, University of Leeds, Leeds LS2 9JT, UK; a.e.beckett@leeds.ac.uk (A.E.B.); edjdsm@leeds.ac.uk (J.M.); 5Leeds Institute of Medical Research, School of Medicine, University of Leeds, Leeds LS2 9JT, UK; rh.uddin1@gmail.com (R.U.); d.p.stark@leeds.ac.uk (D.P.S.); 6Academic Unit of Health Economics, Leeds Institute of Health Sciences, University of Leeds, Leeds LS2 9JT, UK; a.martin1@leeds.ac.uk; 7Teenage and Young Adult Cancer Service, Leeds Teaching Hospitals NHS Foundation Trust, Leeds LS9 7TF, UK; suemorgan@nhs.net; 8Teenage Cancer Trust, London WC1V 7AA, UK; louise.soanes@teenagecancertrust.org

**Keywords:** adolescent, young adult, mental health, emotional wellbeing

## Abstract

**Simple Summary:**

Developing cancer during adolescence or young adulthood adds extra challenges to an already difficult period of development. This can cause a ‘biographical disruption,’ in the sense that people miss out on key milestones they might reasonably have expected to achieve during the period, and this can affect their emotional wellbeing. While research shows that young people may experience various degrees of emotional distress, less is known about the emotional experiences and barriers young people experience when accessing help and support. We interviewed two groups of young people who were within 6 months of diagnosis and 3–5 years after treatment. We found three themes that described emotional health experiences: the emotional impact of cancer; personal barriers to support; and support to improve mental health. Differences were identified across the treatment journey, in which we were able to propose therapies and interventions that could be used to help improve psychological outcomes.

**Abstract:**

The biographical disruption that occurs in adolescents and young adults following a cancer diagnosis can affect various important psychosocial domains including relationships with family and friends, sexual development, vocational and educational trajectories, and physical and emotional wellbeing. While there is evidence of the physical impact of cancer during this period, less is known about the impact on emotional wellbeing and especially on the barriers for young people accessing help and support. We aimed to obtain a more in-depth understanding of young people’s experiences of their diagnosis, treatment, psychological impact, and range of resources they could or wanted to access for their mental health. We conducted an in-depth qualitative study using semi-structured interviews with 43 young people who had developed cancer aged 16 to 39 years and were either within 6 months of diagnosis or 3–5 years after treatment had ended. Framework analysis identified three themes: the emotional impact of cancer (expressed through anxiety, anger, and fear of recurrence); personal barriers to support through avoidance; and support to improve mental health through mental health services or adolescent and young adult treatment teams. We showed the barriers young people have to access care, particularly participant avoidance of support. Interrupting this process to better support young people and provide them with flexible, adaptable, consistent, long-term psychological support has the potential to improve their quality of life and wellbeing.

## 1. Introduction

There is a wealth of information highlighting the unique challenges affecting adolescents and young adults (AYAs) with cancer diagnosis, namely young people aged 15 to 39. Incidence in this age range is rare compared to that in adults, but still more frequent than in children. The increasing levels of survivorship result in additional medical, social, and psychological concerns for this specific group [1].

Historically, there used to be a dichotomy in AYAs’ receipt of healthcare—some being treated in pediatric services, and some in disease-centered adult oncology services. Given the specific biology of AYA cancers, specificity of late effects, as well as age-related developmental differences compared to children and adults, this model resulted in a large level of unmet need [2,3,4]. This led to a motivation to shape the care of AYAs in an age-appropriate manner—namely care services tailored and focused on the developmental particularities posed by the age of the patients, rather than being disease-specific [5].

Internationally, age-appropriate AYA cancer care acknowledges the biographical disruption triggered by a cancer diagnosis given its occurrence at a vulnerable time of multiple developmental transitions—finishing school, going to university, forging new relationships and a career, and building a family [6,7]. The socio-economic and psychological developmental differences between the younger and older halves of this age group are also recognized—namely, the younger will be finishing education or engaging in higher education, whereas the latter will be more likely to worry about their forming or newly formed family [8,9]. For example, currently, in the United Kingdom (UK), AYA services encompass those up to the age of 25 [4,10], while in Canada services encompass a larger age range [11]. However, the concept of ‘emergent adulthood’ highlights the similarity of the turbulence of multiple transitions lived by people within this age range, compared to both younger and older counterparts. Furthermore, given a shift of crucial milestone completions towards later years within newer generations [12], services are progressively being reorganized to include young adults up to the age of 29 or even 39 [4].

AYAs aged 15 to 39 have in common the fact that they are undergoing crucial developmental changes. The shock and uncertainty of a cancer diagnosis cause a ripple effect across most areas of life—biographical disruption [4,7,13,14]—to a larger extent than for older adults with more established social contexts [7,13,14]. This encompasses a wide range of important psychosocial domains such as their relationships with family and friends, their sexual development, their vocational and educational trajectories, and their physical and emotional wellbeing [15]. From the onset of puberty, the adolescent brain undergoes significant neurological development, which continues into their late twenties and early thirties, characterized by synaptic pruning and remodeling [16]. These later ages are encompassed in the concept of emerging adulthood, in which there is a shift towards redefining goals and the future alongside newfound young adult responsibilities [17]. With a diagnosis of cancer young people face a dual crisis in which normative developmental stressors are juxtaposed with the challenges of navigating their journey through treatment [15,17]. Disruptions at these critical developmental transitions have long-reaching chronic effects on quality of life [15]. Much has been written and put into practice concerning advances in young people’s physical treatment and wellbeing, with the rates of survivorship at 5 years post-treatment increasing steadily over the last decade, now approaching 90% depending on the diagnosis [18]. However, the increased length of survivorship does not account for long-term impacts on quality of life and emotional wellbeing [7,8,13,14].

While the impact on survival has been reported, there is less evidence on the impact of cancer on young people’s emotional wellbeing, and even fewer practical interventions have been put in place to better cater to this. Over the last decade, there has been an increase in calls to address the psychological wellbeing of young people with cancer, yet psycho-oncology interventions specifically targeting this age group are sparse and generally exist at more local levels as opposed to national programs [19,20,21]. Globally, AYA cancer care continues to focus primarily on physical health [20]. With the advances in cancer treatments for young people and the increase in those living into older adulthood, it is now critical to better cater to and support the unmet psychosocial needs of this patient group. This is particularly important in a post-pandemic world, which is likely to have impacted them further [3] through concerns related to infection, changes in communication with healthcare professionals, and a shift in the modes of communication and interrelating with peers that may increase feelings of isolation [3,22,23].

Developing psychological support packages to improve the psychological wellbeing, social functioning, and mental health of young people during and after treatment is the top research priority for this population in the UK [24]. It is increasingly understood that cancer seriously impacts young people’s mental health, with studies suggesting greater anxiety and depression in comparison to the general population, and greater psychotropic medication use [25]. Psychological distress is characterized as an unpleasant experience in which psychological, social, physical, relational, physical, spiritual, or sense of self is disturbed [26]. It is at times transient and unavoidable, particularly for those experiencing a critical illness such as cancer [20]. However, psychological interventions can act as a buffer and a balm for unnecessary long-term impacts from distress [27]. There is also a long history of evidence that shows that earlier intervention has much better outcomes for wellbeing in terms of mood and overall quality of life [26] but intervention effectiveness may also be shaped by a host of factors including patient–doctor communication, patients’ age and even ability to narrate their situation [28]. Psycho-oncology interventions have been shown to reduce experiences of anxiety, depression, and distress in cancer patients [29,30]; however, less than 30% of patients with high levels of distress ask for mental health support while on treatment [31,32,33,34,35].

Adolescents and young adults are a diverse group, with variations in age ranging from 15 to 39 years depending on national guidance. Young people encounter a broad spectrum of tumor types or other malignancies, which occur during a critical developmental period [19,36]. Despite this heterogeneity in diagnosis, emotional distress is consistently observed in this population [37,38]. Studies have shown there is an increase in affective disorders at diagnosis which can still be present up to a year later [39,40]. Psychological distress is associated with poorer adherence to treatment and quality of life, even after treatment ends, and the potential for developing further psychological disorders [41,42,43]. A reluctance to engage with support services is frequently noted in cancer patients [44,45], especially in young people, with reports as high as 75% of young people with mental health difficulties in the general public are not in contact with support services [46].

Barriers to accessing support services vary in AYA but often relate to awareness and perceptions of help, cultural and social norms, particularly relating to gender roles, and risks of stigma and rejection [47]. For young people, these barriers may be further compounded by the degree of distress inherent in cancer, and the biographical disruptions inherent to its diagnosis [21,48]. The exact nature of these barriers is still not fully understood [44] but may relate to the type of support that is offered, i.e., 1-to-1 therapies vs. group therapy, in-person vs. online, or a lack of understanding of what psychological support entails [37]. Young people are at risk of not being able to verbalize this distress due to their developmental age and emotional awareness [20,49].

The aim of this study was to obtain a more in-depth understanding of the psychological impact and experiences young people had while they were on treatment and when treatment had ended.

## 2. Materials and Methods

### 2.1. Study Design

This was a mixed method, multi-stage, multi-center study investigating social transitions and reintegration following a cancer diagnosis. This paper reports on the mental health aspects brought up by participants in this sequential qualitative study conducted with young people who had completed a survey. A more in-depth description of all qualitative results is in preparation.

### 2.2. Participants

Young people were recruited from two specialist AYA cancer centers in England to participate in a larger research program—Social Transitions And Reintegration Support (STARS). Following initial contact from the clinical team with the use of a study leaflet and participant information sheet, experienced researchers got in touch with interested participants. The information sheet described the study, which included a survey administered either electronically or on paper twice (at consent and 6 months later) and an optional semi-structured interview (online or via telephone) for which participants could have been selected if interested.

The interviews focused on participants’ thoughts and feelings related to the impact of their diagnosis on their social outcomes (work, education, relationships) and support services they received or would have desired. Here we report the findings related to a specific section of the findings in the qualitative sub-study, while the in-depth results will be published separately, given their depth.

Interested participants were invited to take part in the interview up to 8 weeks following their first or second survey completion to balance the views following a shorter or longer time in the study, while also allowing for the challenges of recruiting busy AYAs into such studies. Additional sampling criteria for inviting interested participants included age, time since diagnosis/treatment, location, and diagnoses. We aimed for a fairly even representation of younger (up to 24) and older (25+) age groups, gender, and between the geographical regions represented by the two cancer centers (North/South). Expecting a change in participants’ views in regard to the perception of their diagnosis, its impact on their lives, and levels of distress throughout time [38,48,50,51], we aimed for an even distribution of patients in the early phase following diagnosis (Cohort 1, recruited up to 6 months post-diagnosis) versus some years following treatment (Cohort 2, recruited 3–5 years post-treatment). Finally, we aimed for heterogeneity in the distribution of the most prevalent and also rarer cancers in this age group.

Participants who were interested in the interview were offered an interview-related information sheet. All participants provided consent to participate and digitally record the interview. The study was approved by the Health Research Authority (reference 20/PR/0428).

### 2.3. Data Collection

Data were collected through semi-structured interviews between November 2021 and January 2023 by four members of the research team (LH, JM, OL, AB). Interviews were conducted by telephone or virtually using Microsoft teams (Version 1.7.00.6058). The interview guide was based on prior psychosocial oncology and sociology research on the impact of a cancer diagnosis on social outcomes and biographical disruption. It aimed to complement the survey with the participants’ personal views of changes in perceptions across time in terms of impact on physical function, education, employment, social life, sources of support, fertility, and emotional wellbeing. It was developed by the research team with support from the young advisory panel—a group of patient experts who shaped the content and offered advice on how it should be delivered to participants (Table 1). Interviews were digitally recorded and transcribed verbatim.

### 2.4. Data Analysis

Data were analyzed using Framework Analysis, which is suited to multiple researchers being involved in the analysis [52]. Framework Analysis comprises five stages:The transcripts were reviewed for accuracy, and the researchers familiarized themselves with the content.A framework was developed deductively based on the interview schedule. This was a means of breaking down the transcripts into multiple working themes and subthemes. For example, the theme “Emotional Impact” included the subthemes: acceptance, anger/frustration, anxiety, autonomy, avoidance, guilt, humor, identity, low mood, overwhelmed, and resilience.The third stage involved indexing each transcript, i.e., reviewing the transcripts and adding the themes and subthemes to them. The framework was not fixed so new themes and subthemes were added as they were identified in transcripts. When new themes or subthemes were added, previous transcripts we re-reviewed to ensure these had not been missed previously.Charting involved developing a matrix within Microsoft Excel (Version 2402), and for each participant, summarizing the text from the transcript, or entering verbatim quotes into the theme (each line in the Excel worksheet was a participant and each column a subtheme).Finally, the chart was examined by the research team to map the themes and subthemes to a higher level of interpretation (as this was a means of deconstructing the transcripts) to gain more insightful meaning into young people’s experiences.

The analysis was led by LH, supported by five members of the research team, and the framework and charting were independently checked by the rest of the team. Themes and subthemes were defined and refined iteratively through multiple consensus discussions within the team, while final charting was pursued by the lead author.

## 3. Results

Forty-three young people participated in the qualitative interviews: 21 within 6 months of diagnosis and 22 were 3–5 years after diagnosis. A total of 11 were aged 16–24 at the time of participation and 32 were aged 25–39. A total of 10 young people had hematological malignancies and 33 oncological. Details on participants’ demographic and clinical characteristics, including cancer severity [53], are summarized in Table 2.

Three main themes were identified related to the psychological impact of cancer: the emotional impact of cancer; personal barriers to support; and support to improve mental health (Figure 1).

### 3.1. The Emotional Impact of Cancer

This theme was represented by three emotions: anxiety, anger, and fear of recurrence.

#### 3.1.1. Anxiety

Anxiety was a common experience for all participants. The shock of diagnosis often came with anxiety for the future, and the liminal spaces between diagnosis and beginning treatment were described as a hard place to be (knowing you have cancer but not starting treatment). There was a lack of support reported during this stage. On-treatment anxiety was expressed as greatly affecting young people day to day and being overwhelmed by anxious thoughts and fears. Some found their anxiety came from poor communication with the clinical team or looking online for more information about their diagnosis, which often led them to catastrophize. While anxiety was to be expected to some degree, some participants really struggled and described having poor mental health. Some felt they had lost confidence in themselves and their ability to handle pressure, feeling worn down, burnt out, and less resilient than they had thought themselves to be. This came with a sense of failure and weakness.

“I think I’ve learned that I’ve got way less mental strength than I thought I had. I thought I was quite a stable person but coming through diagnosis I crumble a lot and I don’t, I really struggle through days, just from basic stuff and I can spiral from really little things,”(Participant 1.162, Cohort 1, male, 25+)

Young people felt they struggled to voice their fears to partners, and friends and consequently were left isolated in their anxiety. Those who were able to communicate their fears openly and honestly had a more positive outlook and were able to work on accepting their circumstances (this was easier for those with a better prognosis). For young people with previous experiences of anxiety, their experience was mixed: some felt they were more prepared to handle anxiety due to previous experiences of psychological support, while others felt totally overwhelmed by the diagnosis. Seeking psychological support was reported as helpful. During treatment, exposure to cancer peers was helpful to normalize their experiences, but it could also be distressing to see how ill people can become, or in other cases to see how ill they were in comparison.

“You’re not sleeping on a night, physically shaking with the anxiety just crying at absolutely everything from—and I’m quite a strong person, quite a resilient person anyway—mentally I’m always like ‘I can face that challenge. Let’s break it down’. But when you hit that brick wall, and it’s not—it’s very much out of the norm for you.”(Participant 1.130, Cohort 1, female, 25+)

Young people who were some years after diagnosis felt anxiety was an entirely unavoidable element of cancer diagnosis and treatment. Many continued to struggle with anxiety after treatment ended; however, noticeable differences emerged in the narrative for those who were able to identify anxiety and seek support during treatment, as opposed to those who avoided seeking support until treatment ended. Participants who reached out for support during treatment commented on how helpful they found having a space to voice their anxieties and fears without it becoming burdensome to those around them and allowing them not to dwell or ruminate on these worries. Fear of death was common, which could leave young people withdrawn into themselves, especially those newly diagnosed. Others felt it was important to understand what triggered their own anxiety, such as reading about cancer online, in order to put boundaries in place.

“I was staring at the ceiling erm oh, honestly it’s, I don’t know, it might have been overwhelming—So, so much was going on erm so much I wasn’t sure of … because I wasn’t sure of what was going on as well, it did really—it-it took a deep effect and it was just like … you’re really empty.”(Participant 2.021, Cohort 2, female, 25+)

Young people who avoided emotional support from professionals while they were on treatment and focused on ‘just getting through’ struggled with anxiety and panic attacks when treatment ended. Some reached out for professional support with varying levels of success due to a lack of accessibility, while others continued to avoid emotional support. While anxiety around self-monitoring health and fear of relapse was common, it was notably more prevalent in those who had not received or sourced emotional support after treatment ended. There were some contradictory narratives with young people reporting they had not experienced anxiety around treatment, yet also expressing high levels of fear and anxiety when it came to relapse and returning for scans and check-ups. Participants who had emotional support either during or after treatment also tended to report having a shift in perspective and resilience, feeling much more able to manage everyday anxiety than they had beforehand.

“Yeah, so like directly after treatment was not a great time. I had like I got panic attacks like quite bad panic attacks like I remember one time specifically I woke up in bed this was like a month after treatment, or maybe it was like right towards the end when we’d had a clear set of scans and was like it was nearing the end of treatment and I woke up in my bed at like 3:00 am shaking just complete body shaking and I did not know why and I could not stop myself…my brain was going like 100 miles an hour and they had to like bring me to hospital…I think I was kind of just scared like deeply petrified.”(Participant 2.619, Cohort 2, female, <25)

#### 3.1.2. Anger

Newly diagnosed young people expressed anger and frustration due to many contributory factors. There was anger most often felt at their general practitioner (primary care physician) and poor handling of their diagnosis. Many participants felt dismissed and ignored and found the initial experiences of diagnosis to be chaotic and upsetting. Along with this, there was anger and frustration in relation to system navigation, particularly with administration staff who made errors in their appointment times, or their communication style was described as rude or difficult to deal with. Some young people felt anger towards their clinical team for poor communication, not feeling listened to, or a perceived lack of support, including difficulty in requesting referrals.

“I went to the doctors and…they basically said it was nothing. It really [expletive] me off if I’m perfectly honest … he sort of just said ‘oh it’s nothing’ but he—I was—I mean—my thing to him was ‘What have you got X-ray eyes? How can you say that it’s nothing?’ Anyway he said ‘well I can refer you but it’s gonna take loads of time because COVID and blah blah blah’. So anyway, he got my back up, so I just said ‘Right. Refer me. I’m not I’m not happy with what you’ve said’. So he referred me and it took like seven months before I had any scans. Because of COVID, every appointment just kept being cancelled.” (Participant 1.159, Cohort 1, male, 25+)

Anger toward friends was also expressed, with reference to their lack of empathy or sympathy, or disengagement. For example, young people found their friends were avoidant of hearing about their cancer treatments or lacking presence. Others also expressed general anger towards cancer and its disruption of their life, struggling to accept their new circumstances.

Young people living with and beyond cancer also recalled the diagnostic period as a source of frustration and anger and felt it had been poorly handled, particularly where they had experiences of their general practitioner being dismissive or unsupportive in pursuing a diagnosis. There was also anger expressed in regard to perceived system failures, such as financial support being unhelpful and difficult to navigate, and errors in administration and treatment.

“I got quite angry, most of the time I was very annoyed.”(Participant 2.315, Cohort 2, male, <25)

Similar to those newly diagnosed, young people felt anger towards friends and family for not providing them with the support they desired, yet they also admitted they themselves had never been able to voice or ask for this support. Young men reported being angry throughout treatment and had not had a way to process or handle their emotions properly, and sometimes took them out on those closest to them, which they now felt guilt over. Young people expressed they still struggled with a lingering sense of anger even after treatment had ended and did not know how to process or manage this emotion.

Furthermore, permanent changes to their bodies and capabilities were a source of anger and frustration for some, particularly things such as being restricted in the long term by having visible changes to their body. These changes, such as scars, served as visible reminders of illness and an emblem of their lack of autonomy in this situation. There was a sense of frustration of having no way to control this change to their body, further compounded by a constant need to manage other people’s commentary and insensitivity in the outside world.

“When I was sick, I would have just loved, just smash up some plates and just … a bunch of anger and then just again, you know in a safe space, obviously. But I think it would just be an interesting sort of thing.”(Participant 2.203, Cohort 2, female, 25+)

#### 3.1.3. Fear of Recurrence

For young people who had finished treatment, fear of recurrence was present for most. Treatment had led them to become more aware of their health and they worried more than previously. The way in which remission was communicated did not feel like a ‘full stop’, which led them to feel ambivalent and anxious that the cancer was not over.

“Since having cancer, I have really bad health anxiety, really bad … Everything kind of settles, then it kind of kicks in a little bit, the mental health side of things. And then it’s like no one’s there anymore … because I think while you’re having your treatment, you’re so busy, you’ve been seen by a medical professional all the time. So, you don’t kind of worry because it’s all in hand. You know, you’ve been seen. But then when everything stops and your treatment stops and then you go home and you’re like’ oh that’s just happened’, and it kind of sinks in then like I wasn’t anxious or overly worried throughout my whole treatment once they said that like you know it hasn’t spread then like I wasn’t worried throughout my treatment then I were happy to just take whatever as it comes it were fine, you know, I’d be fine at end of it. So, whatever, just throw it at me. It’s fine, but it’s the after realization when you’re not as busy and got all these appointments to go to and stuff.”(Participant 2.355, Cohort 2, female, 25+)

Young people whose treatment had ended some years previously continued to live with fear and anxiety of their cancer returning. One difficulty was the importance of participants remaining vigilant of their health and monitoring themselves for the return of symptoms or cancer relapse. However, many felt they experienced symptoms due to anxiety and sat with these uncomfortable feelings for long periods of time. In particular, there was a sense that hyper-vigilance and monitoring of their bodily sensation could cause flights of anxiety and catastrophizing. For many, this fear and anxiety continued. There was also a rhetoric of being healthy and looking after oneself, an almost personal responsibility for cancer not to return and keep their bodies well. While some tried to have patience and self-compassion to allow themselves to accept and adapt to this fear, others were less able to face this.

“After a while it just felt like I’m well for now but am I going to get another cancer diagnosis which will hurt me or will this one come back and hurt me. Or will there be something else and I will die suddenly? So, I say these thoughts are probably a bit intrusive, like, I don’t want them, they just happen. I’ve spoken to one person, like a friend. But I don’t really feel like that’s something people want to hear, maybe that makes me less, happy less able to enjoy my life, or it’s just one of those things. It is what it is.”(Participant 1.113, Cohort 1, female, 25+)

Contact from the clinical team often caused a lot of anxiety, particularly if it was unprovoked and young people had not had regular follow-up. Those on regular follow-up felt less anxious about contact and feeling well monitored seemed to help with their anxiety. Particularly for those who did experience a relapse, it seemed a sense of lingering fear of recurrence was difficult to navigate and very much a shadow in the rhetoric of their recovery.

### 3.2. Personal Barriers to Support

This theme reflected the avoidance behaviors young people exhibited toward acknowledging the emotional impact of cancer and/or seeking help.

Young people who were newly diagnosed reported emotional avoidance, although many were not conscious of this. While some were aware they were avoiding dealing with painful emotions, they could not bring themselves to face them. Most commonly, young people did not want to think about cancer at all, doing anything they could to distract themselves from their anxiety and fears. For some, this was framed as being helpful, while others were aware that it made it harder for them to deal with anxiety and generally caused more distress but had been unsure how to deal with it.

“I’ll be perfectly honest, I’m not someone who requires a lot of support so, you know, I’m quite a private person and whatnot. So if, if someone sort of started trying to, I don’t know, if I were in there, and they were trying to probe me with loads of questions, you know, I’d probably just sort of shut them down anyway because I don’t really want or need much in terms of, you know I’ve got my family. I’ve got my friends. I don’t really need that much in terms of like—like emotional support or anything like that.”(Participant 1.159, Cohort 1, male, 25+)

Some young people found vocalizing any emotion difficult but for others, seeking psychological support was helpful in navigating their fears, which they had been keeping to themselves. They mostly overcame this barrier to support by having someone else, either family, friends, or staff, help them take the step to access support. They found having a space to discuss fears without voicing them to their social support network was helpful. Other young people were more avoidant of psychological support, feeling it would be ineffectual: talking about emotions would not change anything, and psychologists could not know how they felt or that it would make them more upset.

“Suddenly you sort of realize that you that you actually might sort of try to keep it all in, but then by talking about it, you realize that actually is helpful … but if you don’t have to make the first step of call someone or whatever it might be. If it’s just sort of like a chat then maybe that would help some people realize that they can get help if they need it you don’t know, they don’t have to just do it all by themselves.”(Participant 1.084, Cohort 1, male, <25)

This feeling of avoidance was expressed by young people long after treatment ended. Again, avoidance was not always conscious and indeed was sometimes presented as a positive coping mechanism from the point of view of the individual. There were young people who had enough space from treatment to recognize they had done themselves a disservice. A common rhetoric was participants ‘putting their heads down and pushing through’ while trying not to be too engaged in their treatment and its impacts on their life or wellbeing.

“That was one thing that actually did come up, which I now maybe do regret. But at that time I just wasn’t sort of ready for, was just people telling me like you should go talk to someone about this. Like talk to a counsellor or something and throughout treatment and even a bit after treatment finished, I was just absolutely certain that I did not want to talk to any form of like counsellor or therapist or you know psychological support person. I was like oh I’m super fine. I don’t want to talk about it … which was probably very silly and stubborn of me at the time … I don’t know, I’m not sure. I think maybe it was like my own like you know, closed into my shell. Like if I’m, if I don’t talk about it then it means I’m fine and I wanted to believe that I was fine.”(Participant 2.619, Cohort 2, female, <25)

Young men were more likely to refer to feeling they needed to be brave, do it alone, or survive on willpower alone; though some acknowledged this was often due more to stubbornness and a lack of emotional maturity when they had time to reflect on it. Regardless of gender, many participants described purposefully avoiding emotional support, as they were unable to admit to themselves that they needed help or were able to voice these fears to others. Many felt they had purposefully not engaged with cancer treatment information so as not to be overwhelmed. While some maintained this had been helpful, others reflected it had actually made them more anxious. With the ability to reflect on treatment, there were many young people who felt they should have engaged in emotional support when it was offered and regretted their choice to avoid it. Young people who decided to eventually engage with emotional support during treatment identified that it had been helpful and felt like a weight lifted off their shoulders. They identified barriers such as finding it difficult to admit they needed help, difficulty asking for help, and difficulty accessing help. When they had another person to support them in this, they generally found it easier to access support.

Family and friends becoming disengaged was noted to be common, and participants described avoiding sharing their emotions to minimize their own distress. Some wanted to avoid being an emotional burden, while others felt other people often did not give them what they needed, despite not really giving them a chance to do so. On reflection, some participants realized that they themselves had made communication very difficult as they avoided expressing their own needs. Avoidance continued to the end of treatment when they realized they needed support as they were struggling with their mental health and when they accessed help, they found there were significant emotional issues to address. Again, barriers were identified in admitting they needed support with asking for help, and conflict between this and wanting to get on with life and leave cancer behind. Some participants continued to avoid support post-treatment, and often referred to ongoing struggles with anxiety related to relapse, as well as unprocessed emotions such as anger, though did not always make the connection to avoidance of support. Some expressed that when treatment ended, they just wanted to avoid thinking about cancer at all and did their best to pretend it never happened, thus not allowing space to process.

“I’d got myself into a such nasty place—it was just, I was—I felt I was on my own. I wasn’t at all. I wasn’t. Never in this world was I on my own. I had everybody at HOSPITAL, I had all my friends and family. I had my colleagues, I had my bosses. I had absolutely anybody I wanted. I just wouldn’t go to them, just wouldn’t, I was just like No, you don’t—I don’t want anybody to see me like this.”(Participant 2.021, Cohort 2, female, 25+)

### 3.3. Support to Improve Mental Health

Mental health was supported either by specific metal health services or through their AYA treatment team.

#### 3.3.1. Mental Health Services

When it came to mental health support, most young people felt this could be managed better. Those on treatment felt mental health should be addressed with the same importance as their physical health. This was highlighted as an area of improvement, such as having regular mental health check-ins. Participants noted it could take a long time to access mental health support while in treatment, and found they requested it many times before it was provided.

“Meeting with my psychologist, which was really helpful and I really, you know, needed this one in the earlier stage of my treatment? No one offered to me and didn’t support me. They just general called from the nurses, you know and the doctors, but I mentioned that several times that I need to have a psychologist.”(Participant 1.129, Cohort 1, 25+)

Having consistent mental health support from the beginning of diagnosis would have prevented them from feeling stigma, which was itself then a barrier to accessing services later. Having a consistent staff member for emotional support built trust and rapport in the same way they had for their cancer care with a consultant or other members of the clinical team. However, the delivery of mental health support could also be improved. For example, young people noted there were limits to the number of sessions, which was not always helpful nor was being told to go for a walk for your mental wellbeing and building a routine. Participants wanted a more consistent space to voice their worries and fears, rather than a space to problem solve. Those who had previous experienced of mental health support prior to cancer highlighted that having those skills in managing anxiety and mood were helpful in regulating their emotions while on treatment, further highlighting the ways more integrated mental health support could be of use.

“I was in like such a bad place anyway, and I think the main thing that I had was I was very, very angry and I couldn’t really understand what the anger was about because a lot of the time it related things that happened three years before. Uh, so I had already started seeing a therapist, which I think I’m like lucky enough to be able to afford so I’ve done it privately … I’ve managed to keep that up throughout this entire process, and I do think that made a difference. I don’t know how I would be if I hadn’t had that because I see her once a week and she’s been really really good.”(Participant 1.160, Cohort 1, female, 25+)

Young people who were 3–5 years post-diagnosis also reflected that having consistent mental health support imbedded from the beginning of treatment would have been helpful. It was felt that a key barrier to accessing mental health services was being able to admit to oneself that they needed support, as well as being able to ask for it. Had it been treated on par with their physical health, many felt this barrier would have been easier to overcome. Likewise, they felt having regular mental health check-ins would have helped to remove the stigma about accessing support as it would have been a part of treatment from the beginning.

“I think there’s so much support in the medical and physical side of things where I don’t think there is in the mental and emotional support—it isn’t on par with that in regard to its availability … You need to go for physical check-ups and medical check-ups, but you don’t need to go to emotional and mental check-ups kind of thing. So, I think just having a bit more sort of enforced in that respect, because I think it does, it will make people better. And just and just again making it seem like that’s completely fine like…after treatment it’s OK to, you know, have a bit of reaction from any treatment or anything. It’s sometimes seen as not OK … it’s a much bigger deal. If you’ve got depression after treatment kind of thing and they have to treat it as a completely separate thing. Whereas I don’t think it should be. I think it should be again normalized and feel more accessible.”(Participant 2.203, Cohort 2, female, 25+)

Making sure mental health services were easily accessible once requested was key, long waiting lists were unhelpful and added to young people’s distress, particularly if participants had avoided support until they were in a mental health crisis. Participants felt that consistent support would be best while on treatment but having a limited number of short sessions was not enough. At the end of treatment, many young people struggled with their mental health; they felt they had focused so much on physical recovery that they had neglected their emotional wellbeing and now that they suddenly had time to process their experiences, they were extremely overwhelmed. Furthermore, they felt there was little or no mental health support offered at the end of treatment and they had no idea where or how to access it. General mental health services outside of cancer were not suited to their specific needs. Many who were living beyond a cancer diagnosis reflected that having some group and peer support would have been helpful at the end of treatment, even more so than on treatment, as it would bring together a community of people their age, in similar positions to help normalize their experiences. Many expressed struggling with mental health after treatment, particularly feeling isolated and unsure of themselves, and felt there was much more needed in terms of support.

When young people had access to community support groups, these were identified as being critical for emotional, physical, and social recovery. Those who accessed support during treatment felt it was useful as a space to put all their fears and anxieties so as not to burden friends and family, which in turn protected those key relationships. It was expressed by some that even 5 years after finishing treatment there was still a lot of emotion and experiences to address, and that with some distance they were more able to address this in therapy. They felt it had taken them time to get to a place they were willing to explore this more in depth.

“Therapy. Definitely. Having consistent therapy for a while and not just a couple of sessions. I was in therapy for four years and still am because, you need time to get into it and a lot of the time you need a year to, like get over the fact that you don’t wanna do therapy so yeah, like having some kind of contact available for that for that kind of support because it does go on and there are things that come up five years after you treatment related to your experience that you would not even think about two years after. I think having a good support system who have that tangible list of things that need to get done, or things that they need specific things is really helpful because it takes the weight off completely.”(Participant 2.040, Cohort 2, female, 25+)

#### 3.3.2. Adolescent and Young Adult Treatment Team

When asked to identify staff who helped with emotional support, young people both on and off treatments reflected that while many staff played critical roles in supporting them, some were more appropriate than others. For example, Clinical Nurse Specialists were identified as useful for system navigation, medical staff were a wealth of expertise on cancer, and ward nurses were a comforting presence in the hospital. Despite this, participants felt that none of these professionals were trained to provide emotional support and could not be expected to take on this role in addition to their already heavy workload. One role, specific to specialist AYA services did emerge, however, which could potentially be upskilled to provide more emotional support: the youth worker.

Youth workers were referred to positively by young people who were less than 25 at the time of diagnosis. They helped to create an atmosphere that was hospitable and enjoyable on AYA wards. Young people felt it was helpful to speak to them about their wellbeing and different elements of their lives. They felt the youth workers knew them as a person and treated them as individuals. They were the members of the treatment team they could speak to about their emotions. They also facilitated opportunities to engage in social activities and meet other cancer patients in a relaxed atmosphere, which was helpful.

“I did find like just sometimes ranting to the YSC [youth support coordinator], you know was enough in itself. Like and not labelling it like psychological support, more just like having someone you know Because I actually genuinely feel like they care about me, you know? Like, it’s not just a job to them. It’s like they come in and they actually, you know…that that kind of special touch…Like you don’t necessarily need to talk to someone, you just need someone to say like I care about you.”(Participant 1.128, Cohort 1, female, <25)

Young people who were a number of years after treatment noted youth workers had been important for creating warm and hospitable experiences when in hospital, facilitated enjoyable social activities, and navigated young people meeting other young people. This gave participants a sense of normality which was important. Youth workers acted as a form of emotional support outside of the medical team, and young people felt they built good connections with them as a result, as this was what they were there for, rather than it being secondary to their role. In particular, youth workers served to help mediate emotional support with families as well and were good sources of support not just for participants but also for their social support networks as well.

“We had a support worker [youth worker] there whose main job was just to kind of have fun with us and like, chat with us. He wasn’t there to be a particular signpost and that was like one of the best things ever, because you get very used to people who are there to like, care for you, and then sometimes you just want a mate who like it wasn’t your parent… I remember there was a time where like my mum had been sleeping there for ages and I was like, I just wanted a night alone. I don’t want her to sleep here tonight, and my mum was very like, no, I’m not going to leave you alone. And so, it was the support worker who I could talk to and he was like, yeah, maybe she might just want one night of, like, independence. That’s all. It’s nothing to do with you. Like, which was good.”(Participant 2.040, Cohort 2, female, 25+)

## 4. Discussion

This article reports on the findings of a qualitative sub-study embedded in a larger mixed methods study. We pursued an in-depth semi-structured interview with a range of AYAs in two large regional hospitals, who were either up to 6 months post-diagnosis or 3–5 years post-treatment at the time of recruitment into the study. The interviews focused on the perceived impact of a cancer diagnosis on a person’s personal biography and social outcomes—how they saw themselves, the impacts on work/education, relationships, as well as the type of support sought or desired from the time of diagnosis and beyond.

Here were report a subset of our rich qualitative results, which will be detailed further in a separate manuscript. We aimed to ‘look beyond the numbers [54]’, as the mixed methods report, combining the findings of the quantitative and qualitative results, reporting on all aspects of inquiry, is in preparation.

We qualitatively described the emotional impact of a cancer diagnosis during adolescence and young adulthood. Given the expected importance of a timespan perspective on illness representation and its impacts [38,48,50,51], the interviews and their analyses focused on how this manifested before diagnosis and continued throughout the treatment journey until young people were living with and beyond cancer.

All young people in our study presented with some degree of mental health impact, but not necessarily to the pathological level. This supports quantitative data indicating psychological distress presented in 27% of young people [40], a depression diagnosis in 49% [55], and in Norway, the high psychological burden after a cancer diagnosis was reflected in this being a reason young people consulted primary care in 86% of visits [56]. Furthermore, evidence suggests young people with cancer have poorer mental health than their non-cancer peers [57] and poorer mental health was associated with increased healthcare utilization [25]. There is evidence of an increase in mental health concerns in this population following the COVID-19 pandemic [3,22,34].

The impact of cancer at various stages of the journey has been reported previously, especially in relation to emotional distress [38,48,50,51], but with few differences due to age at diagnosis or severity of illness. For example, Forster et al. [58] showed there was an association between prolonged routes to diagnosis and treatment, with poorer quality of life, anxiety, and depression at 6 months post-diagnosis. The end of treatment was noted to be a particularly challenging time for young people, characterized by discordance between how they expected to feel and the reality [59,60]. Similar to our findings, young people found it particularly challenging because access to all the healthcare they had during treatment was perceived as no longer being available or accessible [61]. We also found that those who were 3–5 years after diagnosis exhibited pronounced fear of recurrence. This is well-recognized as one of the most distressing consequences of a cancer diagnosis, prevalent in as high as 97% of patients depending on the cancer type [62,63]. While there is variation in consensus on factors associated with high fear of recurrence [64], age is consistently reported as being related to higher levels, specifically in patients less than 40 years old [64,65]. Unsurprisingly, fear of recurrence is also associated with anxiety and psychological distress [63,64], which may in part explain the mental health impact exhibited in our young people.

While young people in our study reported aspects of the mental health impact shown previously, we noted particularly that anger has been less often reported elsewhere. Poor communication with healthcare providers is reported frequently [66,67,68]; however, anger directed at friends and family for not being engaged or empathetic is an interesting finding and in tension with previous indications young people ‘protect’ their friends and families from emotional burden by not sharing their emotions [69].

Our key finding, however, was the avoidant behaviors young people reported in getting help and support for mental health. Barriers to psychosocial health have been reported previously [70] and this has emerged as one of the biggest unmet needs for this population. Previous health-related behavior research suggests a gendered effect—with males being more likely to adopt an avoidant coping style to healthcare consultations and medical advice [71]. However, avoidant behavior was reported by both males and females in our interviews, in relation to support seeking from formal and informal sources. This points to a tension because young people emotionally want their friends, family, and healthcare professionals to be more empathetic but do not wish to say so, while in order to be empathetic, others need to understand what young people are going through emotionally.

### 4.1. Implications

The experiences of young people newly diagnosed with those 3–5 years after treatment diagnosis create an interesting opportunity in which we might be able to derive some key improvements to young people’s cancer care. We make the following suggestions on how embedding psychological care in teenage, young adult, and even adult cancer treatment may be beneficial from the point of diagnosis, throughout treatment, and into long-term follow-up (Figure 2).

To begin, it is clear there would be benefit from embedding psychological support from the beginning of treatment, rather than using a post-crisis intervention model [72]. Young people in our study reflected that if their emotional wellbeing had been given as much priority as their physical health from the beginning it would have helped them to overcome the barriers of stigma and avoidance which play key roles in their refusal of psychological therapies. There were multiple approaches suggested through which embedding this care might help. For example, many highlighted that diagnosis was a very tumultuous and emotional time and there was often a lack of emotional support during this. Medical staff are more focused on understanding the burden of disease and crafting treatment pathways and perhaps cannot be expected to also provide complex emotional support. Yet participants highlighted the ways in which these experiences impacted them. It is well-documented how difficult the experience of diagnosis can be, particularly when pathways leading up to it have been difficult [58]. The anger and dissatisfaction felt towards their general practitioners in particular was notable. This age group is likely to face difficult paths to diagnosis due to systemic issues within the healthcare diagnostic process, and are unlikely to be resolved quickly, therefore, providing a counterbalance of support once diagnosis has been achieved would be useful.

Participants highlighted that it would have been helpful to have check-in sessions with a mental health professional in the same way in which they had their initial meetings with their medical team. This would not need to be a full psychological session but rather a touch point which also allows for a better building of trust and rapport and working towards destigmatizing the use of supportive services while gaining insights into individual patient needs.

Psycho-oncology programs are often inconsistent with the number of sessions that should be provided to young people [73] and given the heterogeneity of not just cancer diagnoses in this group but also treatment regimens and wider biographical disruptions, a one-size-fits-all formula is unlikely to work. Alternatively, rather than having specific sessions and treatment programs, what might be more beneficial is for patients to have consistent access to onsite low-level mental health support staff. These staff members would be able to provide low-intensity psychological interventions organically, such as psychoeducation, skill building, and mindfulness sessions reflecting the stepped model of care currently used within the UK. AYA nurses are well placed to provide such support and would benefit from further training where appropriate. They are the lynch pin in an AYA service, providing a vital link for the patients on their journey.

Creative approaches to support also have some promising benefits for young people, such as rage rooms to help release pent-up emotion (participant suggestion), and psychological support services would be suited to delivering this. Having a role such as this embedded in the ward team would also help to normalize access and build rapport with patients. Youth workers could provide such support to young people, given their unique presence in NHS AYA wards. These unique posts are charity-funded and have been placed successfully in some hospitals already in the UK and provide low-level support during treatment. They could support patients as they traverse treatment and facilitate important social support activities and peer group support. Should a patient then require higher-tiered support for more disrupted emotional wellbeing, there should be a clear referral to the multidisciplinary team for discussion and referral to clinical psychology or similar, reflecting a stepped model of care [74]. This could (potentially) overcome patient avoidance and normalize accessing psychological support later. Embedding these roles, particularly in young adult and adult cancer care would be helpful as these patients have limited access to both social peer events as well as general emotional wellbeing support. This would help to overcome barriers to accessing support, such as isolation and avoidance.

As a patient progresses through treatment these roles would then also allow for better preparation when it comes to approaching the end of treatment. Our findings indicated this period represents a secondary biographical disruption that was destabilizing for young people, and many felt inadequately prepared for this next step. In particular, those who used avoidance as a coping mechanism reported really struggling with their emotional wellbeing after treatment ended as they then needed to manage with a huge quantity of distress. At this point in time, access to support was significantly reduced, and participants struggled with ongoing feelings of anxiety and trauma. Our evidence indicates that better preparation leading up to the end of treatment would be beneficial, and having group work to bring patients together who are at this similar stage would be helpful. Participants reported that having a cancer peer community was particularly helpful at this time and helped to normalize their struggles to reintegrate while giving them a support base in the process who understood their experiences. Again, psychoeducation and ongoing skills training around managing things such as fear of recurrence would be helpful at this timepoint and there are promising interventions that could be adapted for this [70].

Young people reflected that even 3–5 years after treatment ended, they felt long-term psychological support would be of benefit. Perhaps this would be the best time to introduce specific higher-intensity psychological techniques such as cognitive behavioral therapy or other types of psychotherapy. Participants reflected that continued emotional wellbeing check-ins routinely included in follow-up appointments would be helpful and would help them feel more contained and connected to their treatment teams.

### 4.2. Limitations

The current study had a number of limitations.

It was a self-selected group of young people participating in a wider study, although participation was equally open to all participants, we may have only engaged those who were passionate about this subject or were struggling to access support. All the participants were recruited in hospitals that house specialist AYA cancer services; these are restricted to young people aged 16–24 years but there is the potential that services that are already focused on enhanced supportive care needs could provide services to smaller hospitals that do not. The experiences expressed by our participants could, therefore, underplay the emotional impact of cancer. Evidence suggests that over 60% of mental health disorders present before the age of 25 [75] and we, therefore, may not have captured the effect of biographical disruption occurring at this fragile time of their development, given our participants trended towards young adulthood. Furthermore, while we identified a range of emotional reactions—anxiety, avoidance, fear of recurrence, depression—development of future analyses could incorporate a keener focus on other potential emotional reactions such as disgust or the development of phobias.

Here we reported our qualitative findings on a subset of themes that emerged from our in-depth interviews. The focus of this paper is particularly on mental health and the emotional impact of a cancer diagnosis on AYA patients. This is due to the strength with which this topic emerged for patients across multiple themes and its importance in the post-pandemic world.

Due to the nature of the mixed methods study, which will result in additional publications, we provided no quantification of themes and subthemes—the messages of our participants were considered of utmost importance. A future manuscript describing the full extent of our qualitative findings, integrated with the quantitative data, will provide further insight into potential differences according to cancer and demographic characteristics.

Finally, we aimed to sample and recruit a fairly equal number of patients per age group and geographical region, as well as participants representing all genders. However, we had no representation of participants with genders other than those assigned at birth and we had slightly more participants from the northern cancer center. Our separate quantitative analyses will provide further insights into whether these factors are relevant to the cancer impact and supportive care of the participants.

## 5. Conclusions

We found there were many ways in which we could improve the provision of psychological care to adolescents and young adults with cancer. While much is already described, we have illustrated some barriers young people have to access care in greater depth. The greatest barrier was participant avoidance of support, which often manifested as a narrative of patients keeping their heads down and just trying to get through treatment and dealing with the emotional impacts later. This is a common thread encountered with this population, yet we see evidence of this being a maladaptive coping mechanism for the long term. It is, therefore, crucial to interrupt this process in order to better support young people going through a cancer journey and provide them with flexible, adaptable, consistent, long-term psychological support to improve their quality of life and wellbeing in the near and distant future. This could be done with support from the clinical teams, youth support workers, and allied healthcare professionals who would be in the best position to prime and cue patients to the availability of mental health support at multiple time points throughout their cancer journey. A more directed and consistent approach towards discussing and checking-in on mental health in follow-up medical appointments may also reveal a large set of patients who may be more willing to engage with this type of support after the end of treatment.

What will be important in our wider quantitative study is to explore more broadly the mental health impact according to gender. We noted that young men in particular struggled with feelings of anger and inability to process and express emotions and exhibited avoidance of support. Quantifying this will enable gender-specific support to be developed.

## Figures and Tables

**Figure 1 cancers-16-01332-f001:**
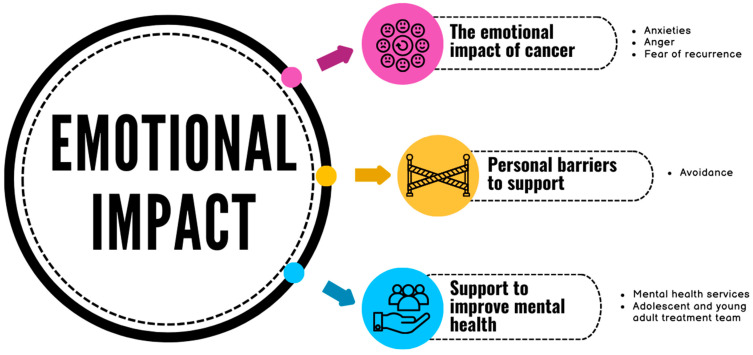
Summary of the themes.

**Figure 2 cancers-16-01332-f002:**
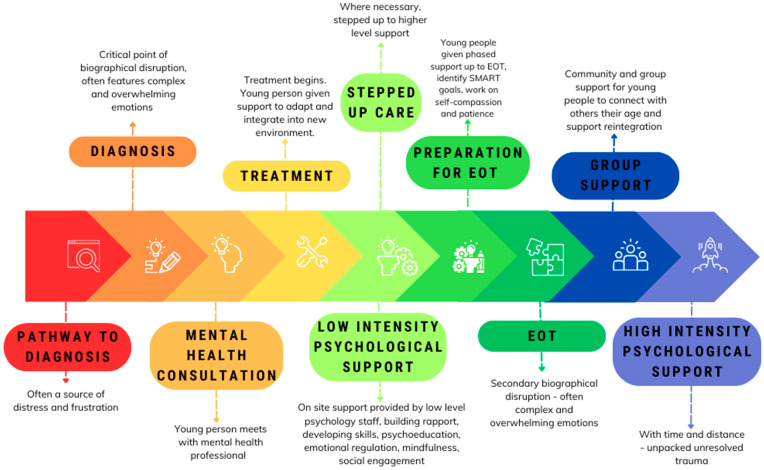
Embedding Psycho-Oncology support for teenagers, young adults, and adults. Note: EOT: end of treatment; SMART: Specific, Measurable, Achievable, Relevant, and Time-Bound.

**Table 1 cancers-16-01332-t001:** Interview topic guide (with guidance for the interviewer).

Topic Checklist
Functional limitations—Attribution and impact on daily activities
Fatigue source/attribution and impact
Education—past and future plans
Reasons for discontinuing education (if discontinued)
Sources of dissatisfaction with education (if in education/training)
Understanding of cancer by others in school (if in education/training)
Support provided by school or others (if in education/training)
Work—past and future plans
Diagnosis disclosure and support from co-workers (if in employment)
Diagnosis disclosure and support from employer (if in employment)
Sources of satisfaction or dissatisfaction with work (if in employment)
Perceived barriers to employment
Perceived pressures to re-enter education or work
Sources of formal and informal support: wanted, available, accessed
* Impact of diagnosis on fertility (if fertility is brought up naturally or the consent to discussing this)
* Impact of fertility on image and life plans (if fertility is brought up naturally or the consent to discussing this)
* Impact of fertility worries on personal relationships (if fertility is brought up naturally or the consent to discussing this)
Taking everything into account, what do you feel like your experiences have taught you—about the world, about yourself, about others

* indicates these were only asked if the explanation (in brackets) applied.

**Table 2 cancers-16-01332-t002:** Demographic and clinical characteristics of patients who participated in the interview sub-study.

	Location	Age	Gender	Time Since Diagnosis	Time Since End of Treatment	Diagnosis	Disease Severity
Cohort 1 ^1^ (n = 21)	North = 14 South = 7	18–36	M = 12 F = 9	1–8 months	N/A	Solid tumors = 12 Hematological = 6	Least = 11 Intermediate or Severe = 10
Cohort 2 ^2^ (n = 22)	North = 11 South = 11	22–38	M = 13 F = 9	3–6 years	3–6 years	Solid tumors = 13 Hematological = 9	Least = 12; Intermediate or Severe = 10

^1^ Up to 6 months post-diagnosis; ^2^ 3–5 years post-treatment. M: male; F: female. N/A: not applicable as a footnote.

## Data Availability

Data are available on request from the authors.

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
