# Peer review of "The Emotional Impact of a Cancer Diagnosis: A Qualitative Study of Adolescent and Young Adult Experience"

_cancers, 2024, doi:10.3390/cancers16071332_

Round 1
Reviewer 1 Report
Comments and Suggestions for Authors
The article entitled, “The emotional impact of a cancer diagnosis: a qualitative study of adolescent and young adult experience” focuses upon a topic of tremendous import, and there is a certainly a need for greater understanding in this area, for the focal age groups identified. Thus, the potential impact of this study is viewed as very high. This study involved a thoughtfully constructed interview topic guide, rich qualitative interview data, and the identification of key themes. This reviewer recognizes the high level of effort that was invested into data analysis and applauds this. The manuscript holds great promise and would benefit from the address of several points as outlined below.
Introduction
· The first paragraph, and elsewhere in the introduction, would benefit from the addition of more recent citations.
· The study identifies adolescents and young adults (AYA) as a heterogeneous group, and it would be helpful to have a greater understanding of why they are combined for the purposes of this study and also the differences that may exist within this broad age range. The justification for the combination of individuals into this large age range should be clear, as it relates to the study of cancer diagnoses. Concurrently, it seems critical to acknowledge the differing developmental processes/issues/services that may exist across this broad age range and/or to contextualize the research based on these developmental differences.
Materials and Method
· Further detail related to recruitment in the Materials & Methods section is needed.
o Who recruited the youth to the study, and how was the study presented – both in terms of the original study as well as the in-depth interview?
o Was parental consent needed & obtained for participants under the age of 18?
o Additional about the original on-line interview sample would be helpful, to contextualize the sample within the overall population of AYA who have been diagnosed with cancer.
o What percentage/number of the larger on-line study participants elected to participate in the in-depth interview? Were there differences between participants in the in-depth interview and non-participants?
· What was the gender of the participants?
· What is the theoretical justification for the two cohort timepoints selected, both in terms of the exact time point as well as the range of time?
· Is there knowledge of when the diagnoses were initially given and/or the length of treatment for those who completed treatment?
· In the interview topic table, there is a qualifying line: “(if fertility is brought up naturally or the consent to discussing this).” Can the authors clarify what this means? Was there much data gathered on this?
· While this reviewer can appreciate that analyses focused upon themes that emerged naturally from the transcripts, one wonders if there is a possibility to further leverage the thoughtfully developed interview, to focus results upon the specific topics that were queried. For example, it would be highly informative to have sections addressing interview topics such as functional limitations in daily activities (including if none are reported), past/future plans for education, and disclosures to co-workers/employers and support received.
· What did the independent checking of the framework and charting involved? For example, did interrater reliability checks or consensus coding take place, to facilitate accuracy of assigned themes and subthemes? How were discrepancies handled?
Results
· It would be helpful to have greater specification of participant demographics by age and cohort. At the minimum, means and standard deviations of age within the two different ranges should be provided. In addition, it would be important to specify the distribution of two different age ranges across the 2 cohorts. Similarly, knowing the types of cancer diagnoses as they relate to the age ranges/cohorts would be helpful.
· A critical concern relates to the need to give readers a greater sense of the number or percentage of individuals that voiced responses that corresponded to a particular theme or subtheme. Terms such as “Many participants” and “Some young people” were viewed by this reviewer as too non-specific to be meaningful. Quantifying the number or percentage of persons with such responses would greatly enhance the paper.
· Relatedly, given the differences in concerns/issues/presentation that may occur between the 2 age groups as well as between the 2 data collection cohorts, having a more consistent sense of who is voicing particular concerns would be very helpful. (When this differentiation did occur when describing results, this provided important context that was appreciated.) This type of information may greatly inform interventions that may be differentially developed for stage of cancer diagnosis/treatment and/or age range.
Discussion
· In general, it was difficult for this reviewer to evaluate the conclusions made in the discussion without more clearly specified results.
· Some additional questions that arose for this reviewer relate to whether the authors took into consideration: differential impact of the type of cancer that was diagnosed; potential impact of the COVID-19 pandemic on their study; differential responses by gender.
· The manuscript would benefit from a careful reading for typos. A comprehensive list will not be provided here, but for example:
o line 98: “effective disorders” is likely meant to be “affective disorders”
o Figure 1: “Person barriers to support” is likely meant to be “Personal barriers to support”
o Line 256: “nowt” is likely meant to be “nothing”
Author Response
Reply to Reviewer 1:
We would like to thank the reviewer for appreciating the work that has gone into this manuscript and their suggestions for improvement. We address their comments below with reference to the tracked changes in our manuscript:
The first paragraph, and elsewhere in the introduction, would benefit from the addition of more recent citations.
As requested, we have included more recent citations related to challenges in offering age-appropriate healthcare to AYAs, AYA models of care, and findings from other qualitative studies in other countries, related specifically to why we’ve decided to run this study.
The study identifies adolescents and young adults (AYA) as a heterogeneous group, and it would be helpful to have a greater understanding of why they are combined for the purposes of this study and also the differences that may exist within this broad age range. The justification for the combination of individuals into this large age range should be clear, as it relates to the study of cancer diagnoses. Concurrently, it seems critical to acknowledge the differing developmental processes/issues/services that may exist across this broad age range and/or to contextualize the research based on these developmental differences.
Internationally, AYA’s care is organised based on the ages of the patients, rather than on the specific disease. We included some clarifications on this in the text. However, while AYA healthcare may be directed to a broad age-range, it’s particularity is that it acknowledges the different developmental processes within this group. We included a new paragraph describing the developmental similarities and differences between the younger and older groups within the AYA age range, as well as in how healthcare is shaped around their differing needs.
Was parental consent needed & obtained for participants under the age of 18?
The age of consent in the UK is 16 so parental consent was not required.
Who recruited the youth to the study, and how was the study presented – both in terms of the original study as well as the in-depth interview?
On page 4 of the manuscript we now include more information related to the methods in this larger research programme and how participants were recruited and sampled to the qualitative sub-study:
Additional about the original on-line interview sample would be helpful, to contextualize the sample within the overall population of AYA who have been diagnosed with cancer.
Table 2 on page 6 now includes a breakdown of the demographic and clinical characteristics of our participants. As we cannot report on cells with a participant number lower than 5 due to risks to confidentiality, we provide an overview of solid tumours and haematological cancers rather than specific diagnoses.
What percentage/number of the larger on-line study participants elected to participate in the in-depth interview? Were there differences between participants in the in-depth interview and non-participants?
The analysis of quantitative results is ongoing. We will report on these differences when integrating the quantitative and qualitative results in our mixed methods report. However, we do not expect major random differences given that we have used a particular sampling method to inviting interested participants into the interview. Not all who presented an interest (nearly all study participants) were selected for the interview both due to theme saturation and sampling strategy.
What was the gender of the participants?
This information is now available in Table 2 on page 6.
What is the theoretical justification for the two cohort timepoints selected, both in terms of the exact time point as well as the range of time?
There is an expectation that illness perceptions, the practical and emotional impact of diagnosis and its treatment will change throughout time, from the point of diagnosis to the end of treatment and even beyond. We have now included this information in the paragraph cited above with appropriate references.
Is there knowledge of when the diagnoses were initially given and/or the length of treatment for those who completed treatment?
We have now included a summary of time since diagnosis or time since treatment (as applicable) in Table 2 on page 6.
In the interview topic table, there is a qualifying line: “(if fertility is brought up naturally or the consent to discussing this).” Can the authors clarify what this means? Was there much data gathered on this?
Table 1 displays the comprehensive topic guide that spans across multiple topics, not all of which would have been applicable to the interviewee. Information in parentheses is guidance for the interviewer, which we now specify in the caption – for instance the interviewer was to ask the participant about their reasons for discontinuing education, if they did discontinue; or to ask about work if in employment, while skipping questions related to education/training if these were not applicable. As fertility is a sensitive topic the guidance to the interviewer was to only touch upon this topic if the participant brought it up naturally or they explicitly consented to this discussion. This was the advice offered by the young advisory panel, who felt that some people may be put off by discussions on this topic. Details on how the topic of fertility (as well as others) were approached by participants will be available in our next manuscript. We now provide more information on how the content of the topic guide was developed on page 4:
While this reviewer can appreciate that analyses focused upon themes that emerged naturally from the transcripts, one wonders if there is a possibility to further leverage the thoughtfully developed interview, to focus results upon the specific topics that were queried. For example, it would be highly informative to have sections addressing interview topics such as functional limitations in daily activities (including if none are reported), past/future plans for education, and disclosures to co-workers/employers and support received.
A longer manuscript describing all our results in depth is in preparation, which will refer to this publication. However, in this publication we wished to offer more space to the topic of mental health in given its importance for this particular group and its presence across multiple areas of our frameworks. We now specify this on pages 4 and 14
What did the independent checking of the framework and charting involve? For example, did interrater reliability checks or consensus coding take place, to facilitate accuracy of assigned themes and subthemes? How were discrepancies handled?
Themes and sub-themes were defined and refined iteratively through multiple discussions within the team. Hence, inter-rater reliability checks would not be applicable as the charting was not performed in a blinded manner, but rather through consensus discussions in the team. This is now specified on page 5 of the manuscript. The benefit of Framework analysis is the transparency in developing themes. The independent review was undertaken by a senior experienced qualitative researcher, who was able to see that the narrative related to each theme was reflected in the content of the transcripts.
It would be helpful to have greater specification of participant demographics by age and cohort. At the minimum, means and standard deviations of age within the two different ranges should be provided. In addition, it would be important to specify the distribution of two different age ranges across the 2 cohorts. Similarly, knowing the types of cancer diagnoses as they relate to the age ranges/cohorts would be helpful.
This information is now available in Table 2.
A critical concern relates to the need to give readers a greater sense of the number or percentage of individuals that voiced responses that corresponded to a particular theme or subtheme. Terms such as “Many participants” and “Some young people” were viewed by this reviewer as too non-specific to be meaningful. Quantifying the number or percentage of persons with such responses would greatly enhance the paper. Relatedly, given the differences in concerns/issues/presentation that may occur between the 2 age groups as well as between the 2 data collection cohorts, having a more consistent sense of who is voicing particular concerns would be very helpful. (When this differentiation did occur when describing results, this provided important context that was appreciated.) This type of information may greatly inform interventions that may be differentially developed for stage of cancer diagnosis/treatment and/or age range.
Our approach was interpretivist. This means that we were interested in understanding the subjective meanings and experiences of our participants, within a social context. According to such an approach, reality is constructed and interpreted by individuals differently, according to their experiences. We are sceptical of attempts to quantify qualitative data because "when working within an epistemic framework underpinned by subjectivity and multiple realities, quantification can actually serve to undermine more complex and nuanced interpretations of the meaning of qualitative data, thereby undermining the quality of our qualitative work." (Monrouxe and Rees, 2019: 187). We employed Framework Analysis. This is one of the most structured forms of qualitative data analysis. The themes that we report are 'interpretive concepts or propositions that describe or explain aspects of the data' and 'are the final output of the analysis of the whole dataset' (Gale et al, 2013). Whilst Framework Analysis is systematic and produces a matrix - and can thus be very appealing to researchers more used to quantitative research - Gale et al (2013) warn against 'the temptation to quantify qualitative data (e.g. “13 out of 20 participants said X)'. Why? because, 'this kind of statement is clearly meaningless because the sampling in qualitative research is not designed to be representative of a wider population, but purposive to capture diversity around a phenomenon'. We were interested in understanding different outcomes for different patients and what processes might be impacting upon patient outcomes.
We have included details of the cohort gender and age for each quote. The quotes are used to illustrate the theme so it is unclear what inferences the reader would get from knowing the severity of disease or cancer type, we have therefore not included this detail.
In general, it was difficult for this reviewer to evaluate the conclusions made in the discussion without more clearly specified results.
We have explained our approach to analysis and interpretation in the response above. This has not been identified as an issue with the other reviewers and therefore we have not made any changes. This qualitative work is a sub-study within a mixed methods research project; hence the desire was – at this point in the work – to look ‘beyond numbers’. However, the qualitative work will be integrated with the quantitative work in future publications. We now specify this in the discussion on page 14.
Some additional questions that arose for this reviewer relate to whether the authors took into consideration: differential impact of the type of cancer that was diagnosed; potential impact of the COVID-19 pandemic on their study; differential responses by gender.
Thank you for this comment. To our knowledge, there is little differences between cancer types and genders in the levels of emotional distress reported by AYA patients – however, our quantitative data may be able to suggest more in this sense. We now specify this lack of difference from previous studies on page 14. Generally, given that AYA are a specific population which is rather rare compared to other older age groups, subgroup analyses on gender and cancer types is not always possible. Particularly, it would not be feasible for this qualitative study due to the low number of participants overall posing a risk to patient confidentiality. We now included in our discussion references to responses by gender and the pandemic on pages 14 and 15.
The manuscript would benefit from a careful reading for typos. A comprehensive list will not be provided here, but for example: line 98: “effective disorders” is likely meant to be “affective disorders”, Figure 1: “Person barriers to support” is likely meant to be “Personal barriers to support”, Line 256: “nowt” is likely meant to be “nothing”
Thank you for pointing that out. Any such typos, including the ones specified, have now been corrected.
Reviewer 2 Report
Comments and Suggestions for Authors
Authors conducted an in-depth qualitative study using semi-structured interviews with 43 young people who had developed cancer aged 16 to 39 years and were either within six months of diagnosis or 3-5 years after treatment had ended.
The article is well written, easy to read.
I cannot justify the methods as qualitative research has other methodology than quantitative studies. However I feel that authors should modify the abstract. They wrote a long background but only short result part. They should shorten the background, defining the study aim, and describe results in more details.
Author Response
I cannot justify the methods as qualitative research has other methodology than quantitative studies. However, I feel that authors should modify the abstract. They wrote a long background but only short result part. They should shorten the background, defining the study aim, and describe results in more details.
We would like to thank this reviewer for their positive comments regarding the readability of our article. In reply to their comment, the abstract now includes more information on the aim of our manuscript and the findings of our research: Given the nature of qualitative research, however, we cannot go into more details on any of these findings without the abstract becoming too verbose. Below are the changes we made to our abstract:
Abstract: The biographical disruption which occurs in adolescents and young adults following a cancer diagnosis can affect various important psychosocial domains including relationships with family and friends, sexual development, vocational and educational trajectories, and physical and emotional wellbeing. While there is evidence of the physical impact of cancer during this period, less is known about the impact on emotional wellbeing and especially on the barriers for young people accessing help and support. We aimed to get a more in-depth understanding of young people’s experiences of their diagnosis, treatment, their psychological impact and range of resources they could or wanted to access for their mental health. We conducted an in-depth qualitative study using semi-structured interviews with 43 young people who had developed cancer aged 16 to 39 years and were either within six months of diagnosis or 3-5 years after treatment had ended. Framework analysis identified three themes: the emotional impact of cancer (expressed through anxiety, anger, and fear of recurrence); personal barriers to support through avoidance; and support to improve mental health through mental health services or adolescent and young adult treatment teams. We showed the barriers young people have to access care, particularly participant avoidance of support. Interrupting this process to better support young people and provide them with flexible, adaptable, consistent, long-term psychological support has the potential to improve their quality of life and wellbeing.
Reviewer 3 Report
Comments and Suggestions for Authors
The authors present a sequential qualitative study (framework analysis) that aims to get a more in-depth understanding of the psychological impact and experiences young people had while they were on treatment and when treatment had ended. The authors conducted an in-depth qualitative study using semi-structured interviews with 43 young 39 people who had developed cancer aged 16 to 39 years and were either within six months of diagnosis or 3-5 years after treatment had ended. Framework analysis identified three themes: the emotional impact of cancer; personal barriers to support; and support to improve mental health.
I would like to raise the following concerns.
Authors are suggested to provide more detailed demographic or clinical data, such as cancer types and diagnostic information, in Table 1.
The severity of cancer may also influence the emotional impact of cancer on adolescents and young adults, as well as their personal barriers to support and the support available to improve mental health, including its psychological impact.
Author Response
Reply to Reviewer 3
Authors are suggested to provide more detailed demographic or clinical data, such as cancer types and diagnostic information, in Table 1. The severity of cancer may also influence the emotional impact of cancer on adolescents and young adults, as well as their personal barriers to support and the support available to improve mental health, including its psychological impact.
We would like to thank the reviewer for this suggestion. Table 2 on page 4 now includes a breakdown of the demographic and clinical information related to the participants, including cancer severity.
Reviewer 4 Report
Comments and Suggestions for Authors
The topic is important and the qualitative approach is a valid point of view to narrate the experiences of AYA cancer.
There are only some minor issues to address to improve the paper.
In the introduction I suggest adding some studies that adopted narratives or qualitative methods, i.e. Tremolada M., Bonichini S., Basso G., Pillon M. (2018). AYA cancer survivors narrate their stories: predictive model of their personal growth and their follow-up acceptance, European Journal of Cancer Nursing, 36, 119-128. https://doi.org/10.1016/j.ejon.2018.09.001.
I suggest stressing also the post pandemic period especially in the relationship with medical staff (find quotation) and to improve communication also adopting social networks or specific web portals. i.e. you can quote: Tremolada M., Taverna L., Vietina F., Incardona R.M., Pierobon M. Bonichini S., Biffi A., Bisogno G. (2023). Adolescents and young adults with oncohematological disease: Use of social networks, impact of SARS-COV-2 and psychosocial well-being, Frontiers in Psychiatry, DOI 0.3389/fpsyt.2023.1239131
Methods
How is it decided the plan of topics of the interviews?
The data narratives are assigned to the different topics, but it isn’t explained which procedure is followed in these assignment. Is there assessed an inter-rater agreement between the researchers that identified themes and subthemes?
I think that many other emotions could be identified such as general fear not only of recurrence or anxiety has different natures (general, performance, phobia…). Also disgust could be for example a possible emotion. This could be added as a research recommendation in the discussion section.
It could be useful to have frequencies of the recurrence of themes/subthemes throughout the different participants to understand which one could be more frequent for example.
Discussion
Possible clinical recommendations on psycho-social needs and supports could be added in the discussion.
Author Response
Reply to Reviewer 4
In the introduction I suggest adding some studies that adopted narratives or qualitative methods, i.e. Tremolada M., Bonichini S., Basso G., Pillon M. (2018). I suggest stressing also the post pandemic period especially in the relationship with medical staff (find quotation) and to improve communication also adopting social networks or specific web portals. i.e. you can quote: Tremolada M., Taverna L., Vietina F., Incardona R.M., Pierobon M. Bonichini S., Biffi A., Bisogno G. (2023).
We thank the reviewer for their suggestions for improvement. We have now included additional recent references including the narrative approach in the publication they suggested. We also now comment on the potential influence of the COVID pandemic.
How is it decided the plan of topics of the interviews?
Thank you for this comment. We now included additional detail on how the interview topic guide was developed, on page 4.
The data narratives are assigned to the different topics, but it isn’t explained which procedure is followed in these assignment. Is there assessed an inter-rater agreement between the researchers that identified themes and subthemes?
Themes and sub-themes were defined and refined iteratively through multiple discussions within the team. Hence, inter-rater reliability checks would not be applicable as the charting was not performed in a blinded manner, but rather through consensus discussions in the team. This is now specified on page 5 of the manuscript:
I think that many other emotions could be identified such as general fear not only of recurrence or anxiety has different natures (general, performance, phobia…). Also disgust could be for example a possible emotion. This could be added as a research recommendation in the discussion section.
Thank you – we now specify this on page 17.
It could be useful to have frequencies of the recurrence of themes/subthemes throughout the different participants to understand which one could be more frequent for example.
Thank you for this comment, which mirrors some of the comments also received from Reviewer 1. A longer manuscript describing all our results in depth is in preparation, which will refer to this publication. However, in this publication we wished to offer more space to the topic of mental health in given its importance for this particular group and its presence across multiple areas of our frameworks. We now specify this on page 4 and 14.
We also refer to this in our Limitations section on page 17, which re-iterates our desire to not rely on quantifying themes and sub-themes in this piece of work, but that this will be done as part of the mixed methods integration of our quantitative and qualitative findings.
Possible clinical recommendations on psycho-social needs and supports could be added in the discussion.
We added a paragraph related to this point on page 18 of our manuscript.
Round 2
Reviewer 1 Report
Comments and Suggestions for Authors
The authors provided an extremely thorough and thoughtful response to comments. Enthusiasm for the study is very high!
Reviewer 3 Report
Comments and Suggestions for Authors
It seems there is a typographical error. It should be Table 2 instead of Table 1. The correct sentence would be: 'Table 2. Demographic and clinical characteristics of patients who participated in the interview sub-study severity.'